# Aging in Place: Connections, Relationships, Social Participation and Social Support in the Face of Crisis Situations

**DOI:** 10.3390/ijerph192416623

**Published:** 2022-12-10

**Authors:** Sacramento Pinazo-Hernandis, Mauricio Blanco-Molina, Raúl Ortega-Moreno

**Affiliations:** 1Social Psychology Department, Faculty of Psychology, University of Valencia, 46010 Valencia, Spain; 2School of Psychology, Faculty of Social Science, National University, Heredia 40101, Costa Rica

**Keywords:** COVID-19, stress, emotional well-being, aging in place, proactive strategies, socio-emotional resources

## Abstract

Objectives: We seek to identify active coping strategies used by older adults to face the pandemic and to deal with daily stressors, and to clarify which factors had an effect on stress, positive emotions and depression in active and healthy community-dwelling older adults in the first and second year of the pandemic in Costa Rica. Methods: Participants were living in their own homes in Costa Rica (*n* = 218, mean age 69.96, 82.1% women). Participants were interviewed by phone and answered an online survey, which included socio-demographic information, mental health variables such as stressors (perceived health and fear of COVID-19, illness, perception of pandemic gravity), loneliness (whether they felt lonely and how often they felt lonely), access to Information and Communication Technologies, socio-emotional coping variables, social participation and physical activity level during the pandemic. Results: Positive socio-emotional indicators related to well-being such as self-efficacy, social support, perceived health and proactive behavior were high. Negative well-being indicators such as perceived stress, emotional COVID-19 fear and loneliness showed low values in the sample studied during both years. We found significant relations across the dependent variables (perceived stress, positive emotions and depression) by studying the psychological well-being coping strategies. Conclusions: Findings highlight the importance of coping strategies and social participation in the capacity of older adults to mitigate the negative psychological consequences of crisis situations and provide evidence of “aging in place”.

## 1. Introduction 

### 1.1. Aging in Place during COVID-19 Pandemic

Aging in place means that people continue to live in their own home and environment for as long as they are able and interested in doing so. The scientific literature has found different motivations that influence peoples’ desire to age at home: be autonomous, have independence to be able to choose freely what to do at any given moment, familiarity with the environment and be able to perform activities of daily living [1]. 

Older adults prefer to age in their own home for as long as possible. Aging at home is related to feelings of attachment to their home and social connections to the community, sense of security and a feeling of autonomy. It reflects and reinforces a sense of attachment to the home and the neighborhood, with improved well-being and social connectedness as a result [2]. 

One of the key factors to aging at home is the fit between the person and their surroundings. Aging at home allows older adults to retain, maintain and nurture their social and family networks. It is a concept that refers to older adults living independently at home with the use of care and support services to meet their needs. Participation in the community and in neighborhood life, and social support are essential to successful aging at home. The World Health Organization defines healthy aging as the process of promoting and maintaining functional ability (determined by the individual’s intrinsic capacity (physical, mental and psychosocial) in the environment (including physical, social and policy environment and their interaction)) to enable well-being in older age [3]. Wiles et al. [4,5] stated that aging in place is related to a sense of identity both through independence and autonomy and through relationships and solidarity roles in the places where people live. Participation in lifelong learning programs, volunteerism and political, social and civic organizations enhances overall well-being in later life [5]. Programs that encompass a wide range of services, such as physical exercise, Information and Communication Technologies (ICT) solutions and education, play a key role in active and healthy aging.

The COVID-19 pandemic drastically altered neighborhood life and social participation. Many resources that are usually key sources of socialization, support, care and activity for older adults were permanently closed for many months and when they were opened they operated with restrictions. It is likely that so many months of avoiding crowded places, being isolated at home and using only online services will have profound consequences for older adults' daily behaviors and well-being in the near future [6,7,8,9,10].

During the COVID-19 pandemic, access to social connections and interactions were limited to reduce transmission risk, especially to older adults. As Smith et al. [11] described, these strategies protected older adults but at the same time increased the risk of social isolation and mental health problems [12,13], such as loneliness, depression, stress, anxiety and reduced social support [14,15,16,17,18]. The threat to mental health and its consequences has caused concern for this group during the pandemic [19,20,21,22,23].

The impacts of the COVID-19 pandemic must be addressed by understanding the multidimensional vulnerabilities of previous inequalities in socio-economic status, gender and age [24]. At the beginning of the COVID-19 pandemic, older adults with health problems had a higher probability of worsening or dying from related complications due to the uncertainty of the appropriate sanitary measures, poor understanding of the disease behavior and unavailable vaccination [25]. The pandemic also raises the alarm regarding the existing gaps in the health systems for older adults [26,27]. 

### 1.2. The Context Studied

The health emergency was declared in Costa Rica on 15 March 2020, after having confirmed the first case of COVID-19 on the 7th of the same month. Costa Rica has public institutions focused on health implemented throughout the territory, through Basic Comprehensive Health Care teams responsible for primary care, clinics and general and specialized hospitals. The governing body for health in the country is the Ministry of Health which, in coordination with the National Commission for Risk Prevention and Emergency Care, the governing body for emergencies, and the Presidency of the Republic, who assumes command of the Emergency Operations Committee, together with the participation of representatives from other institutions in the country, organized the response to the pandemic threats and consequences. 

The solid public institutional framework of the country, considering ministries, autonomous institutions and other institutions and highlighting the role of local governments and public universities, made it possible to address health needs in a comprehensive manner, with the challenge of attending not only to the impact of the pandemic on health and economy, but also to handle this systemic emergency in a country with a high population and territorial inequality. Among the actions to highlight, access to vaccination for the population in vulnerable conditions is presented since the end of 2020, having applied, to date, at least the first vaccine to more than 87% of the country’s population, with 95% of people between 12 and 100 years or older being vaccinated [28].

### 1.3. Theoretical Frameworks

Older adults may strengthen the ability to deal with adversity even in the pandemic context [29]. Pearman et al. [30] indicate that anxiety towards COVID-19 was associated with COVID-19 stress, but proactive coping was associated with less COVID-19 stress. The pandemic context has increased loneliness feelings regarding physical distancing [31,32]; changes in everyday activities in older adults could predict better or worse functioning, health and psychological well-being. 

Kahana’s Preventive and Corrective Proactivity model (PCP) is a model that incorporates actions that individuals can make to adapt to age-related changes and stressors, and to optimize their health and quality of life [33]. The PCP model proposes that internal and external resources may facilitate behavioral adaptations that contribute to better health [34].

ICT use is one of the proactive behaviors. Older adults with a higher level of digital literacy were able to work through more emotions due to operationally effective ICT use. During the pandemic and because of restrictions on going outdoors and contact with social relations, ICT use has become more important, showing the need to reduce the digital divide.

The support provided by ICT to older adults had already been demonstrated before the pandemic, but it has been more evident in this moment of crisis and in times of social distancing where emotional support has been of great help, as it has allowed older adults to feel connected to their loved ones [35,36,37].

### 1.4. Purpose of Study

This study aims to deepen the knowledge of the strengths of older adults by taking data at two points in time one year apart (2020 and 2021) and thus overcome the limits of other interesting studies that were conducted only in the first months of the pandemic, such as Finlay et al. [6] whose data were collected during the summer of 2020.

This study has the principal aim to analyze coping strategies and resources that older adults put in place as psychological responses to cope with the crisis generated by the COVID-19 pandemic, as an example of healthy aging trajectories and the aging-in-place agenda. Taking into account that there is not a formula for dealing with the current COVID-19 pandemic, identifying the various strategies used is mandatory [38,39]. Research evidence could relativize the vulnerability tag associated with old age, as a key element to work against the extended ageism culture [40,41]. 

The present study was based on the hypothesis that the appropriate coping strategies that older adults put in place during the COVID-19 pandemic were key to their well-being. Heterogeneity in old age highlights the importance of studying different groups of older adults in pandemic context-dependent research.

The aim of the present study was to explore the phenomenon of aging in place in people who age successfully. Research on the experience of older adults in times of pandemic will contribute to a better understanding and will enable policies and interventions aimed at promoting well-being in other future times of crisis.

## 2. Materials and Methods 

### 2.1. Design

The study followed a quantitative longitudinal panel design. We followed the same people in two evaluation times: one in 2020, during October and November (*n* = 218). Then, we ran follow-up during July and August of the year 2021 (*n* = 180, 83% answer rate), when in Costa Rica the COVID-19 vaccine campaign began around the country. 

### 2.2. Sample and Participants

The inclusion criteria of the study were participants had to be 60 years or older, living in Costa Rica during the years 2020 and 2021 and be community-dwelling active students in university educational programs for older adults (U3A), or in community groups for older adults at senior centers during the years 2019 and 2020 just before the pandemic. We selected older adults with a profile of active social participation lifestyles, such as those who participate in U3A or community groups, because of their evidence of healthy aging trajectories in past studies in the Costa Rican context regarding the PCP model [42]; we also wanted to gain evidence of the PCP during a context that was highly stressful such as the COVID-19 pandemic. We invited participants through these community groups and educational programs around the seven political regions of the country.

Participants answered a self-administered questionnaire, which was about 30 min long, and were directed to an online survey platform, hosted by Qualtrics (Provo, UT, USA), which included socio-demographic information, perceived health and mental health variables such as stress, depression and loneliness; we also measured the emotional effects of the pandemic with situational fear and perception of the gravity of the pandemic. We measured coping variables such as social support, self-efficacy, resilience, positive emotions and proactive behaviors (social participation, physical activity). The number of older adults who participated in the study in 2020 was 218, and 180 participated in the follow-up in the year 2021. The socio-demographic variables shown were collected in 2020. The people studied reported an average age of 69.96 years with a range of 60–90 years. Almost all were women (82.1%). Most of them lived with family members (78.4%). More than half of the sample were retired (61.5%), with a high level of study reached (51.9%). In terms of the sample’s COVID-19 disease exposition, during time 1 (2020) 1.5% of the sample reported getting infected by COVID-19 and during time 2 (2021) 97.2% reported having had a full COVID-19 vaccination scheme. The Costa Rican National Council of Research in Health gave its ethics approval, and those attending the research completed electronic informed consent. 

### 2.3. Measures

#### 2.3.1. Socio-Demographic Control Variables

−Age: number of years;−Education: years of formal education; −Income: average monthly income. 

#### 2.3.2. Stressors

−Perceived health: Measured with two items, asking participants would you say your health is ‘poor’ (1) to ‘excellent’ (5) and about their current physical health in comparison to their pre-pandemic level using the same scale. High scores indicated better health (α = 0.88).−Illnesses: With a list of self-report chronic illnesses, we measured the presence or the absence of the illness. We created a dummy variable of (0) absence, (1) presence.−Vaccine: Having had the COVID-19 vaccine, yes or no. We created a dummy variable of (0) absence, (1) presence.−Emotional fear of COVID-19: To indicate the severity of the fear of COVID-19 we used four items from the fear of COVID-19 scale (FCV-19S) developed by Ahorsu et al. [43]. Items ask participants how much they are afraid of COVID-19: if they are afraid of losing their life because of it, if they feel uncomfortable to think about it and if they become nervous when watching the news about it. Each item is measured with a 5-point scale (range values = 1 to 5). Higher scores indexed greater emotional fear about COVID-19 (α = 0.86).−Pandemic concern perception: We used 4 items with a 5-point Likert scale to measure the perception of concern of the pandemic at personal, familiar, community and country level. A higher score indicates a worse perception.

#### 2.3.3. Internal and External Resources

−Perceived social support: We used 5-item items from the Social Network Scale [44] to represent two scores (parcels) which were indicators of satisfaction with family and with friends. Items used a 5-point scale (1 = not satisfied to 5 = greatly satisfied). Higher scores indicate higher levels of satisfaction with support. (α = 0.78 and 0.76.)−Functional social support: In order to measure social support, we used the 11-item scale of the Functional Social Support Questionnaire DUKE-UNC [45] to measure the degree of perceived social support [46]. A higher score indicates a better perception of social support (α = 0.91).−Self-efficacy beliefs: We used a 5-item scale that measured the ability of individuals to cope with difficult situations, have perseverance and have a positive outlook about the future. Items are answered using a 5-point Likert scale (range = 5 to 25) where each response was coded from 1 ‘never’ to 5 ‘always’. Higher scores indicated greater self-efficacy (α = 0.78).−Resilience: We used the Spanish version [47] of the Resilience Scale (RS-14) [48]. It measures the degree of individual resilience, considering a possible personality characteristic that allows coping with adverse actions. Items were answered using a 5-point Likert scale (14–70). Higher scores indicated better resilience (α = 0.88).−Loneliness: We used 2 items adapted from Losada-Baltar et al. [32] to measure perceived loneliness during the pandemic. Participants were asked if they felt lonely and how often they felt lonely. Items used a 5-point scale (range = 2 to 10), where each response was coded from 1 ‘hardly ever feel lonely’ to 5 ‘always feel lonely’. Higher scores indicated greater loneliness (α = 0.92).−Proactive healthy strategies: We used 3 items to measure healthy effective coping strategies related to exercising (walking, home workouts), cognitive activities (reading, learning, writing) and healthy eating. Items used a 4-point scale; respondents reported the increase in each proactive behavior since COVID-19 started. Participants indicated how often they engage in such activities (range = 3 to 12), using a scale from 1 ‘never’ to 4 ‘always’. Higher scores indicate a greater use of healthy strategies (α = 0.58).−Physical activity and social participation: We used the Spanish version of the Physical Activity Questionnaire for Older Adults, CHAMPS [49]. We selected 15 items to measure the frequency of physical activity and social activity, using a scale from 1 ‘never’ to 5 ‘7 days a week’. Higher scores indicate a higher frequency of activities (α = 0.67).−Information and Communication Technology—ICT use: With an index of three items we measured ICT use, such as mobile and apps (e.g., WhatsApp, Facebook and similar) for communication and leisure. Higher scores indicate a higher frequency of use. 

#### 2.3.4. Dependent Variables

−Perceived stress: We used a self-report 6-item scale in which participants rated the frequency of experiencing negative emotions since the beginning of COVID-19. These items are adapted from the Cohen et al. [50] stress scale. Respondents rated the influence of the pandemic on how they feel impacted by unexpected events, nervous, stressed, not able to carry on, despondent because they have no control over events or feel that they cannot overcome many difficulties. Items used a 5-point scale (1 = none to 5 = most of the time). A higher score indicated perceived stress (α = 0.79).−Positive Affect and Negative Affect Scale (PANAS) of Watson, Clark and Tellegen (1988) [51]: The scale seeks to measure emotional instability as an initial tool for assessing emotional state. It has 20 items (10 positive emotions and 10 negative emotions). Items used a 5-point scale (1 = never to 5 = very). A higher score in the Positive Emotion sub-scale indicates a higher presence of positive emotions. A higher score in the Negative Emotion sub-scale indicates a higher presence of negative emotions. (PANAS P, α = 0.87, PANAS N, α = 0.89.)−Depression (Center for Epidemiologic Studies Depression Scale, CES-D), ref. [52]: We used a short adapted CES-D version. The CES-10 Scale [53] is a self-report depression scale for research in older adults which includes 10 items. Items used a 4-point scale (1= rarely to 4 = all the time). A higher score indicates depression (α = 0.76).

### 2.4. Statistical Analysis

Our analysis uses structural equation modeling with latent variables (SEM) to estimate the prediction effects of stressors (health, emotional fear of COVID-19), external resources (social support), internal resources (self-efficacy) and proactive activities (healthy behaviors) on loneliness and perceived stress while controlling for socio-demographic characteristics during the first year of the pandemic. Our operational model (Figure 1) uses seven latent variables with their corresponding indicators/dimensions and two observed variables (age and education). 

We used Amos 26 version to estimate the model with a maximum likelihood (ML) procedure. To reduce the number of parameters estimated, we executed a parceling technique to model the latent construct “social support”. We created two parcels representing composite indexes instead of using five separate items of social support. Parceling is appropriate since we were interested in examining the nature of a set of constructs rather than knowing the specific relations among the items [54]. 

To declare the model fit acceptable, we used the criteria of the Comparative Fit Index (CFI) and the Tucker–Lewis Index (TLI) above 0.90, and the Root Mean Square Error of Approximation (RMSEA) below 0.08 [55].

We also ran a hierarchical linear regression analysis to explore the prediction effects over two dependent variables (positive emotions and depression) during the second year of the pandemic. We used SPSS 26 version to estimate the models. 

## 3. Results

In this section, we present the analyses made as follows: First, the descriptive data from the variables measured by time of evaluation, T1-2020 and T2-2021, are shown. Then, we present the SEM PCP model tested in T1-2020 and finally the hierarchical linear regression analyses made by the data of T2-2021. 

### 3.1. Description of the Variable Studied

As it is shown in Table 1, positive indicators related to well-being such as self-efficacy, social support, perceived health and proactive healthy strategies showed high values. Negative well-being indicators such as perceived stress, emotional COVID-19 fear and loneliness showed low values in the sample studied during T1-2020. 

As it is shown in Table 2, in T2-2021 positive indicators related to well-being such as resilience, functional social support and positive emotions showed high values. Negative well-being indicators such as perceived stress and depression showed low values. ICT use was high, and physical activity and social participation presented medium scores. The score of the perceived health indicator was low which is evidence of better health perception; finally, the average of reported chronic illness was low too.

### 3.2. SEM Analysis PCP Model Tested

Following Kahana’s corrective and proactive model, the SEM model includes six exogenous latent variables representing stressors, resources and proactive factors affecting the dependent variable “perceived stress”. We also included two socio-demographic control variables: age and education. Figure 2 displays the estimated standardized beta coefficients for each variable in the model. The SEM model fitted the data well as indicated by the Comparative Fit Index (CFI = 0.925), the Tucker–Lewis Index (TLI = 0.902) and the Root Mean Square Error of Approximation (RMSEA = 0.047, low 0.037 and high 0.057). The chi-squared value was 339.99, *p* < 0.05. The percentage of variance explained by the SEM model on the dependent latent variable, perceived stress, was R^2^ = 0.553.

Significant (*p* < 05) direct effects on the dependent variable were detected from three latent constructs: fear of COVID-19 (beta = 0.37), self-efficacy (beta = −0.26) and loneliness (beta = 0.023). A higher emotional fear of COVID-19 and loneliness were associated with increased perceived stress, and higher self-efficacy beliefs attenuate perceived stress. 

There were significant indirect effects on stress level involving stressor variables, such as perceived health (−0.54), which negatively influence proactive behaviors and in turn attenuate loneliness (−0.18). Another significant indirect effect on the level of stress comes from social support, a mediator latent construct. Social support impacts negatively on loneliness (beta = −0.34) and positively on self-efficacy (beta = 0.38). Self-efficacy contributes positively to proactive behaviors (beta = 0.22) and in turn attenuates loneliness and perceived stress (−0.26). Fear of COVID-19 also has an indirect significant effect on stress through its impact on loneliness (beta = 0.24).

### 3.3. Hierarchical Linear Regression Analysis 

We made two linear regression models to test the PCP model in the T2-2021 sample. Both models looked to test the predictive effects of the socio-demographic variables (age, education, income) as the first step in the regression model. Then, the second step was the health variables (perceived health, chronic illness) as stressors, and the third step was the pandemic context variables (perceived stress, pandemic concern and vaccine), also as stressors. As the fourth step we included the proactive variables (ICT use, physical activity, social activity level), and finally as the fifth step we included the internal and external resources (functional social support, resilience). (See Table 3.) 

The first model put as the dependent variables the mean of positive emotions. In this model, the best predictor was one proactive variable, the physical activity level (beta = 0.17, *p* < 0.01), which means that more physical activity levels contribute to more positive emotions. There was one internal resource, resilience (beta = 0.33, *p* < 0.01), and one external resource, functional social support (beta = 30, *p* < 0.01), which means that more resilience and social support contribute to more positive emotions. The percentage of variance explained by the model on the dependent variable was R2 = 44%. 

In the second model, the dependent variable was the mean of depression. The best predictors were the stress level (beta = 39, *p* < 0.01), which means that more stress contributes to more depression, two proactive variables, ICT use (beta = −13, *p* < 0.05) and social activity (beta = −0.16, *p* < 0.01), which means that more ICT use and social activity contribute to less depression and, finally, more functional social support contributes to less depression, as an external resource (beta = −23, *p* < 0.05). The percentage of variance explained by the model on the dependent variable was R^2^ = 58%.

## 4. Discussion 

The primary aim of this work was to provide an approximation of the determinants of well-being on older adults during pandemic times. The descriptive data show that we are dealing with a group of successfully aging individuals. You may recall that this is a group of people involved in U3A and community active groups. In T1-2021, self-efficacy, social support, perceived health and proactive behaviors were high. Negative indicators of well-being such as perceived stress, emotional fear of COVID-19 and loneliness showed low values. In the scientific literature generated during this period, older adults tend to report less loneliness, depression and anxiety symptoms compared with younger people. These outcomes indicate that older adults must have used better strategies to cope with stressors during the COVID-19 pandemic [32,56,57,58].

In T2-2022, the sample is characterized by high levels of social and physical activity: ICT use, resilience, positive emotions and functional social support, few chronic diseases or illnesses, low levels of depression and low perceived stress. In addition, high values of proactive behaviors stand out. 

Participation in learning and social activities does have an impact on adults’ levels of life satisfaction, an important aspect of subjective well-being [59,60], and has positive effects on those personal attributes that are most closely associated with resilience [61,62] and seem to protect individuals against depression [63,64]. The descriptive data show the same findings.

In the PCP SEM model (with an explained variance of R^2^ = 55%), the dependent variable was perceived stress. Significant direct effects were detected on the dependent variables of fear of COVID-19, self-efficacy and perceived loneliness. On the other hand, a greater emotional fear of COVID-19 and greater perceived loneliness are related to greater stress and, on the other hand, greater self-efficacy beliefs attenuate perceived stress. 

With respect to indirect effects on the level of stress, we evaluated two of them: perceived health and loneliness. Perceived health influences proactive behaviors and attenuates perceived loneliness. Social support negatively influences perceived loneliness and positively influences self-efficacy. 

In the first regression model (with an explained variance of R^2^ = 44%), the dependent variable was positive emotions. Significant positive effects were detected on the dependent variable of physical activity as a proactive behavior and on the internal and external resources of social support and resilience. 

An interesting element of Kahana's model that has served as our framework is proactive behaviors. ICT use, social activity and psychical activity showed high values and were directly related to positive emotions.

The impact of the pandemic has been very large across all population groups. However, the best predictor of positive emotions were proactive variables (physical activity and ICT use) and internal and external resources (resilience and functional social support), meaning that the higher the levels the more positive emotions and less depression. 

With respect to depression, in the second model with a high explained variance (R^2^ = 58%), the best predictors were stress level, which means that more stress contributes to more depression, and two proactive variables, ICT use and social activity, which means that more ICT use and social activity contribute to less depression, and finally as an external resource, more functional social support contributes to less depression. 

It has been posted that older adults have great abilities to deal with adversity. For example, Aspinwall and Taylor [65] called these actions *proactive coping*, which is when the person makes an effort to respond to potential stressors either to prevent them or to modify their form before they occur. Older adults and young adults tend to appraise situations differently, so the subjective assessment and emotion-attribution of a situation is an important factor, since the same situation could be rated as more, less or not stressful for some individuals in specific contexts or socio-demographic conditions [66]. 

In both T1 and T2, perceived stress was low. This finding is related to approaches that describe emotional well-being at old age, derived from the socio-emotional selectivity theory [67]. This theory indicates that the perception of a reduced number of stressors during the aging process is related to an individual and motivational effort in shifting towards positive experiences and emotionally meaningful activities. Neubauer, Smyth and Sliwinski [68] stated that the reduced amount of stress reported by older adults might also be explained by the development in older ages of a higher threshold to consider a negative event as a stressor.

The feeling of belonging to a neighborhood and the community is very relevant over the course of life and it is manifested through social participation and active engagement in the U3A and senior centers, where the participants can create social clues. Participating in social and educational activities is a determinant factor to feeling less lonely, being involved in more social participation and engaging in more practices of healthy aging. Being less engaged in social activities increases the likelihood of older people moving away from their houses [69].

Participation is key in the lives of these older adults. The analyses have again highlighted the role of social support and how those with productive aging behaviors, who exercised and already had some ICT management and were able to stay in touch with their relations during the pandemic, suffered less loneliness, had less stress and greater well-being over time. The use of ICT by older adults mitigates their loneliness and social isolation and facilitates digital interactions that connect them to the outside world, improving their well-being [70,71].

The unexpected global pandemic situation has forced governments to take drastic measures that have proven to have negative results. Reflection on what has happened to community-dwelling older adults and analyses such as these can help us to be more prepared for new crises. 

## 5. Conclusions

Aging in place is a desire of most older adults. Home and neighborhood conditions, social services support and networks are determinants in the possibility of aging at home. Sufficient and appropriate community services are needed to support healthy aging at home, but in a context of crisis, such as the one that we have been living through, good social and health care coordination has been even more necessary, especially to buffer the stress experienced.

The participants of the present study were engaged in different learning and social activities that increased their feelings of belonging to the community and peer relationships and enabled them to better fight health problems and loneliness. 

The promotion of older adults’ autonomy and well-being, together with the creation of an active network of health and social services, may improve the possibility of older adults aging at home [71].

Interesting research conducted in the times of the pandemic has been showing factors that threatened the well-being of older adults, such as Finlay et al.’s [5] research. However, there are few longitudinal studies that show the impact of this and even fewer that focus on the capabilities of successfully aging people living at home. Limited research has examined this. This paper fills an important gap by exploring older adults’ experiences during two years of the COVID-19 pandemic, using a longitudinal perspective. 

The data are intended to shed light on the impact of the pandemic on people aging successfully in Costa Rica. An online questionnaire was completed through the internet; therefore, certain internet skills and having a computer and Wi-Fi connection at home were necessary. Some of the participants were older adults involved in U3A and community active groups, so they were a group of successfully aging individuals who were young–old with an average age of 69.96 years, with a high level of study. The group does not correspond to the total population and does not represent the vulnerable population of older adults. The data should be taken with caution, but although they are not all like this, there is a growing group of aging people with average educational levels with more tools for self-care and skilled internet users.

The data are generalizable to similar people but unfortunately many other people in social exclusion, with economic problems, in a situation of dependency, illness or poverty or without ICT management could not face the pandemic in the same way. Notwithstanding these limitations, a key contribution of this research comes from the longitudinal data allowing us to learn about the advantages of the benefits of successful aging and aging in place. More health care behaviors, knowledge and management of new technologies and social support have been shown to be key to better coping with the pandemic in this group of older adults.

Communities should prepare for potential multi-crisis contexts (economic, health, climate, etc.) in the future and take advantage of the fact that many people have diverse capacities to use proactive strategies on their own and support them in doing so. The use of ICT in education and health has only just begun and the pandemic has highlighted the need to reduce the digital divide and take advantage of all of its possibilities. However, we cannot forget that many people cannot enjoy successful and healthy aging because they are in multiple contexts of exclusion due to disability, economic problems, living in places at risk of depopulation, etc. Public policies must take this into account to strengthen services that allow people to age better and where they most want to: in their own homes.

Finally, a methodological limitation of the present study is the cross-sectional analytical approach of the two time periods measured. It will be important for future analysis of the sample to take into account the attrition effects in longitudinal hypothesis testing.

## Figures and Tables

**Figure 1 ijerph-19-16623-f001:**
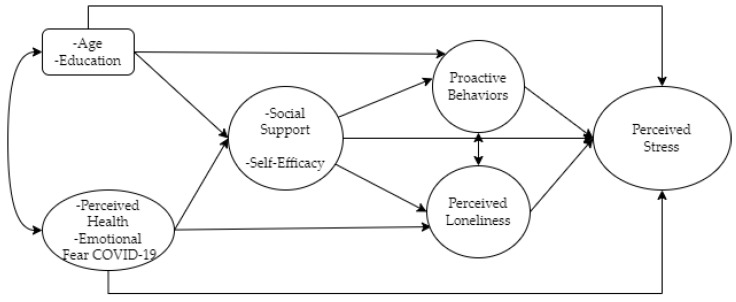
Operational PCP model tested.

**Figure 2 ijerph-19-16623-f002:**
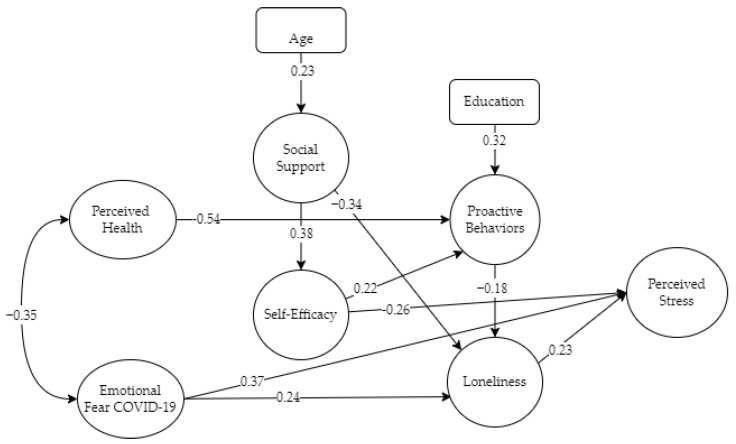
SEM model tested.

**Table 1 ijerph-19-16623-t001:** Description of the variables analyzed in T1-2020.

Variable	*n*	M (SD)	95%CI
Perceived Stress	215	2.27 (0.74)	[2.16–2.37]
Loneliness	215	1.73 (1.13)	[1.58–1.88]
Emotional COVID-19 Fear	215	2.04 (0.99)	[1.90–2.17]
Perceived Health	212	4.02 (0.78)	[3.91–4.12]
Self-Efficacy	216	4.46 (0.61)	[4.37–4.54]
Proactive Healthy Strategies	216	2.76 (0.60)	[4.37–4.54]
Social Support	218	4.10 (0.84)	[2.67–2.84]

Sourc: CI = Confidence Interval.

**Table 2 ijerph-19-16623-t002:** Description of the variables analyzed in T2-2021.

	*n*	M (SD)	95%CI
Illness	180	1.92 (1.48)	[1.69, 2.13]
Pandemic Concern Perception	180	4.21 (0.84)	[4.08, 4.33]
Perceived Health	179	2.23 (0.91)	[2.09, 2.36]
Functional Social Support	179	3.97 (0.94)	[3.83, 4.11]
ICT Use	179	4.08 (1.17)	[3.90, 4.24]
Resilience	179	6.33 (0.67)	[6.23, 6.43]
Perceived Stress	179	2.20 (0.65)	[2.10, 2.29]
Physical Activity	179	2.18 (0.79)	[2.06, 2.30]
Social Participation	179	2.07 (0.51)	[1.99, 2.14]
Positive Emotions	179	4.20 (0.66)	[4.10, 4.40]
Depression	179	1.63 (0.51)	[1.55, 1.70]

Source: CI = Confidence Interval.

**Table 3 ijerph-19-16623-t003:** Hierarchical Linear Regression Models for T2-2021.

Independent Variables		Dependent Variables	
		Model I	Model II
		Positive Emotions	Depression
Steps		Beta	Beta
1	Age	−0.01	0.05
	Education	−0.09	0.01
	Income	−0.07	−0.15 **
∆R-squared %		0.01	0.06
2	Perceived Health	−0.05	0.16 *
	Illnesses	0.02	0.14 *
∆R-squared %		0.05	0.34
	Perceived Stress	−11	0.39 **
3	Pandemic Concern Perception	−0.01	−0.03
	Vaccine	0.01	−0.08
∆R-squared %		0.08	0.13
4	ICT Use	0.12	−0.13 *
	Physical Activity	0.17 **	−0.02
	Social Participation	0.06	−0.16 **
∆R-squared %		0.09	0.02
5	Functional Social Support	0.30 **	−0.23 *
	Resilience	0.33 **	−0.02
∆R-squared %		0.21	0.04
*F*-Test Value		*F* = 11.75 **	*F* = 20.12 **
R-squared %		44%	58%

Source: * *p* < 0.05, ** *p* < 0.01. Beta: all are standardized regression coefficients. *n* = 180.

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
