# Peer review of "Aging in Place: Connections, Relationships, Social Participation and Social Support in the Face of Crisis Situations"

_ijerph, 2022, doi:10.3390/ijerph192416623_

Round 1

Reviewer 1 Report

The article has an interesting and topical purpose, with an important impact on society. However, it has certain shortcomings that need to be addressed:

1.  In certain parts of the text, the words aging and ageing are used interchangeably. I recommend using the same word to refer to this term.

2. In the introduction, more references are needed to support the content described. For example, in lines 39 or 46.

3. The terminology ICT or WHO is used from the outset without explaining what it refers to.

4. In the control variables, why is it not taken into account whether the persons assessed were living with someone or not? This is a very important variable in terms of the coping strategies used.

5. Different age groups are mentioned because of the heterogeneity that exists in this population. Why is this not mentioned in the introduction?

6. Why is information collected from the age of 60 onwards? Explain the reason for this minimum age.

7. In the purpose of the study it is stated that information is collected at two different points in time (2021 and 2022) but in the design it is stated that it was in 2020 and 2021. This needs to be clarified.

8. When information is given about the age and sex of the participants, it is not specified whether it is from the sample of the first measurement or the second. This is necessary as there was experimental loss.

9. Nothing is described about the experimental loss, nor are attrition calculations made to see if this could have affected the analyses in any way.

10. There is a need to establish inclusion/exclusion criteria for the study. 

11. Why was it not measured whether people had a psychological disorder?

12. Was it controlled whether participants had had covid or experienced any traumatic situations due to covid? This is very important as it can be a powerful extraneous variable.

Author Response

Manuscript ID  IJERPH-2025210

Dear IJERPH editors

First of all, we would like to thank you for the time dedicated to the review and the interesting contributions that help us to improve the article. Indeed, we could have better explained the article's strengths and limitations.

In this new version, we hope to have responded to the reviewers' suggestions.

Reviewer 2 Report

First of all, we would like to congratulate and thank the authors for their excellent work, especially in the methodology section, for the quantity and quality of the tools used in the study. Undoubtedly, a longitudinal study was necessary to show how social and community participation can help older adults in crisis situations.

However, it seems important to me to point out aspects that may bias the results, and which should therefore be added to the discussion and limitations section, such as the fact of university studies, and the place where the sample was collected. It would be necessary in the discussion section to talk about this issue, whether people with more educational resources not only know more health tools, but also more ICT tools. Another aspect related to this is the cognitive reserve capacity achieved thanks to this. I would like this topic to appear as well.

Finally, it would also be necessary to include a section on the strengths of the study, where the authors explain whether these results could be extrapolated to other future crisis situations, even if they are not related to the pandemic.

Best regards

Author Response

Manuscript ID  IJERPH-2025210

Dear IJERPH editors

First of all, we would like to thank you for the time dedicated to the review and the interesting contributions that help us to improve the article. Indeed, we could have better explained the article's strengths and limitations.

In this new version we hope to have responded to the reviewers' suggestions.

Reviewer 1

However, it seems important to me to point out aspects that may bias the results, and which should therefore be added to the discussion and limitations section, such as the fact of university studies, and the place where the sample was collected. It would be necessary in the discussion section to talk about this issue, whether people with more educational resources not only know more health tools, but also more ICT tools. Another aspect related to this is the cognitive reserve capacity achieved thanks to this. I would like this topic to appear as well.

Finally, it would also be necessary to include a section on the strengths of the study, where the authors explain whether these results could be extrapolated to other future crisis situations, even if they are not related to the pandemic.

Authors answer:

We appreciate the reviewer's comments to improve the manuscript. We added new information regarding the reviewer’s recommendations in the discussion and conclusion section
